# Ferroptosis in Renal Cancer Therapy: A Narrative Review of Drug Candidates

**DOI:** 10.3390/cancers16183131

**Published:** 2024-09-11

**Authors:** Lingyan Yu, Yuyueyang Qiu, Xiangmin Tong

**Affiliations:** 1Zhejiang Chinese Medical University, Hangzhou 310053, China; 2Laboratory Medicine Center, Department of Clinical Laboratory, Zhejiang Provincial People’s Hospital (Affiliated People’s Hospital), Hangzhou Medical College, Hangzhou 310014, China; 3Department of Biology, Grinnell College, Grinnell, IA 50112, USA; 4Department of Laboratory Medicine, Affiliated Hangzhou First People’s Hospital, School of Medicine, Westlake University, Hangzhou 310006, China

**Keywords:** ferroptosis, renal cancer, herbal extract medicine, natural compound, nanomaterials, clinical trials

## Abstract

**Simple Summary:**

Treatment options for patients with advanced renal cancer are limited, as one of the main methods, chemotherapy, is prone to drug resistance in the course of treatment. There have been numerous studies confirming that ferroptosis is involved in the development of renal cancer and drug resistance during treatment. In this review, we provide new insights into the resistance of classical chemotherapy drugs sorafenib and sunitinib in renal cancer and explore the mechanism of different types of ferroptosis-related drugs to treat renal cancer, such as herb extract medicine, ferroptosis inducers, natural compounds, nanomaterials, etc. We have also conducted an in-depth discussion on the application of ferroptosis-related drugs in the clinical treatment of renal cancer.

**Abstract:**

Renal cancer is a common and serious malignant tumor of the urinary system. While surgery effectively treats early-stage renal cancer, advanced cases pose a significant challenge due to poor treatment outcomes and chemotherapy resistance. Therefore, there is an urgent need to develop alternative therapeutic strategies. Ferroptosis is a newly defined form of programmed cell death characterized by the accumulation of iron-dependent lipid peroxides, which plays a critical role in tumor progression and drug resistance. Recent studies have shown that ferroptosis is involved in the occurrence and development of renal cancer, and ferroptosis-related genes can induce cell apoptosis and can be used as potential biomarkers for early diagnosis of renal cancer and participate in drug resistance of renal cancer chemotherapy. With the continuous improvement of the mechanism of ferroptosis, drugs targeting ferroptosis for the treatment of renal cancer are emerging in an endless stream. Based on the theoretical basis of the occurrence of ferroptosis, this paper reviewed drug-induced ferroptosis in renal cancer cells from the aspects of herbal medicine, natural compounds, drug resistance mechanisms, and nanomaterials, and delves into the clinical application potential of ferroptosis-related drugs in the treatment of renal cancer.

## 1. Introduction

Renal cancer is a disease with a rising global incidence, with 4,348,490 new cases and 155,953 deaths worldwide as of 2022 [1], and studies predict that the incidence will continue to increase over the next decade [2]. Due to its insidious and asymptomatic nature, renal cancer is often diagnosed at an advanced or metastatic stage, with a five-year survival rate of less than 20% for these patients (accounting for 30% of cases) [3]. Unlike early localized kidney cancer, which can be treated surgically, advanced and metastatic renal cancer is primarily managed with immunotherapy, targeted drugs, and chemotherapy [4]. Despite the effectiveness of current treatments to some extent, therapeutic options remain limited, making it crucial to research and develop new antitumor strategies. Interestingly, recent studies have found that renal cancer is sensitive to ferroptosis [5].

Ferroptosis is a form of cell death distinct from traditional apoptosis, driven by the accumulation of lipid peroxidation products and lethal reactive oxygen species (ROS) resulting from iron metabolism [6]. Over the past decade, research has revealed that ferroptosis involves complex biological processes triggered by iron metabolism, lipid peroxidation accumulation, and antioxidant imbalance. With advancements in understanding the mechanisms of tumor ferroptosis regulation, substantial evidence suggests that inducing ferroptosis in tumor cells holds promise as a new therapeutic approach for cancer treatment [7,8,9,10]. This paper reviews the induction of ferroptosis in renal cancer cells from the perspectives of traditional Chinese medicine, natural compounds, resistance mechanisms, combination therapies, nanomaterials, and clinical trials providing new strategies for ferroptosis-based therapy.

## 2. Mechanisms of Ferroptosis Regulation

Ferroptosis is a kind of programmed cell death mode driven by three mechanisms: iron metabolism abnormality, lipid peroxide accumulation, and antioxidant system imbalance (Figure 1).

Iron-dependent programmed cell death is fundamentally regulated by the metabolism of iron ions, the equilibrium of the antioxidant system, and the key reaction of lipid peroxidation. Together, these processes dictate the cell’s sensitivity and resistance to oxidative stress.

### 2.1. Iron Metabolism Abnormalities

Iron metabolism is a crucial cellular process in the occurrence of ferroptosis. Research indicates that an overload of intracellular iron can lead to ferroptosis in tumor cells [11]. In the human body, iron typically exists in two forms: ferric ion (Fe^3^⁺) and ferrous ion (Fe^2^⁺). Ferric ions bind with transferrin and interact with transferrin receptor 1 (TFR1) to enter the cell [12]. Once inside, ferric ions are reduced to ferrous ions by the ferric reductase STEAP3. These ferrous ions preferentially form various iron-binding complexes involved in numerous physiological and biochemical reactions. When these complexes approach saturation, excess ferrous ions accumulate within the cell, forming a labile iron pool. Additionally, excess iron can be stored as ferritin. Ferritin heavy chain possesses ferroxidase activity, converting ferrous ions back to ferric ions and safely encapsulating them within the ferritin shell to reduce the levels of free iron.

Thus, intracellular iron is primarily stored in two states: as ferritin or as free ferrous ions in the labile iron pool. Under normal circumstances, the intracellular iron concentration remains dynamically stable. However, during ferroptosis, the excess ferrous ions in the labile iron pool participate excessively in the Fenton reaction, generating reactive oxygen species (ROS), such as hydroxyl radicals. The accumulation of ROS leads to lipid peroxidation of the cell membrane, resulting in cellular dysfunction and death.

### 2.2. Abnormal Accumulation of Lipid Peroxides

The excessive accumulation of lipid peroxides is a critical trigger for ferroptosis. The regulation of lipid peroxidation within cells is a highly sophisticated system. Currently, two primary processes for intracellular lipid peroxidation are known. One process involves the enzyme-catalyzed peroxidation of lipids: polyunsaturated fatty acids (PUFAs) are converted into highly reactive lipid peroxides through the catalytic action of various enzymes. Studies have shown that the main sources of these PUFAs include not only the cell membrane system but also arachidonic acid (AA) and linoleic acid, which are widely present within cells [13].

The other process of lipid peroxidation within cells is mediated by free iron ions through the Fenton reaction. During ferroptosis, the excessive free ferrous ions undergo the Fenton reaction, producing ROS that reacts with the PUFAs in the lipid membrane to generate lipid peroxides. Subsequently, these lipid peroxides interact with ferrous ions to produce peroxyl radicals, which can extract hydrogen from adjacent acyl groups in the lipid membrane environment, thereby propagating the lipid peroxidation process [14].

### 2.3. Antioxidant Metabolism

Iron accumulation and lipid peroxidation within cells are the two central biochemicals tightly controlled by the cellular antioxidant defense system. Glutathione peroxidase 4 (GPX4) is a crucial antioxidant enzyme first identified in the ferroptosis process. Its mechanism involves eliminating hydroperoxides in the lipid bilayer and preventing the accumulation of lethal lipid ROS [15]. In addition to the GPX4 pathway, three other mechanisms have been discovered to be involved in ferroptosis regulation: the FSP1/CoQ10/NADPH pathway [16], the DHODH pathway [17] and the GCH1/BH4 pathway [18]. These pathways primarily regulate intracellular reducing agents like GSH and CoQ10, thereby reducing ROS levels to control lipid peroxidation and ferroptosis.

Moreover, a recently reported amino acid oxidase, interleukin-4-induced-1 (IL4i1), has also been found to clear free radicals and regulate genes associated with ferroptosis inhibition, thereby suppressing ferroptosis [19]. This represents an emerging potential approach for ferroptosis regulation.

## 3. Ferroptosis and Renal Cancer

Renal cancer has become a prominent threat to human life. Histologically, the majority (90%) of renal cancer cases are renal-cell carcinoma (RCC), which mainly includes clear-cell renal-cell carcinoma (ccRCC; 70%), papillary renal-cell carcinoma (pRCC; 10–15%), and chromophobe renal-cell carcinoma (ChRCC; 5%) [20]. To explore the deep connection between ferroptosis and tumors, studies have used erastin—a classical ferroptosis inducer—on 60 tumor cell lines from eight tissues, revealing that RCC cells are more susceptible to erastin-induced ferroptosis. Further investigation showed that this induction is accompanied by ROS accumulation and decreased GPX4 expression, and these effects can be reversed by antioxidants [21]. This suggests that erastin-induced cell death in RCC is closely related to ferroptosis.

Additionally, based on data from The Cancer Genome Atlas (TCGA) and the Genotype-Tissue Expression (GTEx) database, the expression of key ferroptosis regulators such as GPX4, SLC7A11, and FSP1 was found to be significantly upregulated, while ACSL4 expression was notably downregulated in the three major types of RCC [22]. In studies of the most common type, ccRCC, silencing GPX4 was found to reduce GSH synthesis, induce lipid peroxidation, and significantly decrease ccRCC cell number [21]. Furthermore, ferroptosis inducers like erastin, BSO, sulfasalazine, and sorafenib can directly or indirectly promote GSH depletion, inducing ferroptosis and thereby inhibiting RCC progression. SLC7A11 also plays an important role in RCC development; both p53 and BRCA1-associated protein 1 (BAP1) can inhibit SLC7A11 expression, thus promoting ferroptosis to suppress RCC development [23,24,25].

ChRCC cells contain high levels of glutathione and GSSG and exhibit higher sensitivity to ferroptosis inducers [26]. In hereditary leiomyomatosis and renal-cell cancer (HLRCC), inactivation of fumarate hydratase (FH) leads to significant accumulation of fumarate [27], resulting in extensive protein acidification, reduced GPX4 activity, and increased susceptibility to ferroptosis. Researchers have also found that the density of RCC cells affects their sensitivity to ferroptosis via the transcription regulator taz-mediated epithelial membrane protein 1 (EMP1)-NOX4 pathway [28].

Therapeutically, sorafenib has been approved by the FDA for the treatment of metastatic and advanced RCC [29]. It is not only a tyrosine kinase inhibitor but also an inducer of ferroptosis in certain cancer cells. In addition to sorafenib, other ferroptosis inducers, such as artesunate and salinomycin, have been identified and developed for renal cancer treatment [30,31].

Moving beyond the exclusive use of ferroptosis inducer, further research also suggested the great potential of combining ferroptosis induction treatment and chemotherapy as a strategy to overcome drug resistance [32,33]. Moreover, key ferroptosis factors GPX4 and ACSL4 are closely associated with multiple tumor-related signaling pathways, including tumor proliferation, EMT, angiogenesis, and tumor inflammation pathways [34,35,36,37,38]. The expression of various ferroptosis-related genes can accurately predict the prognosis and survival outcomes of patients, serving as potential prognostic biomarkers and therapeutic targets [39].

These findings collectively indicate that ferroptosis is involved in the initiation, progression, and metastasis of renal cancer. Compared to normal renal cells, cancer cells exhibit higher sensitivity to ferroptosis [40]. Targeting ferroptosis can reduce damage to normal cells during treatment and alleviate drug resistance in cancer cells [11]. Thus, we believe that ferroptosis-targeted therapies for renal cancer can better guide personalized and precise treatment. Future research should continue to explore more ferroptosis-targeted therapeutic methods and drugs for renal cancer.

## 4. Current Status of Drug Research Targeting Ferroptosis in Renal Cancer

In the current field of research on ferroptosis-targeted drugs for renal cancer, we pay special attention to several key categories of drugs: herbal extract medicine, natural compounds, novel nanomaterials, and ferroptosis inducers. These drugs induce ferroptosis in renal cancer through different experimental conditions (Table 1) and mechanisms of action (Figure 2). At the same time, these drugs are significant for the treatment of renal cancer, overcoming chemotherapy drug resistance, and the development of new therapeutic strategies in combination with immunotherapy.

### 4.1. Traditional Chinese Medicine and Natural Compounds in Treating Renal Cancer

#### 4.1.1. Icariin II

Icariin II (ICS II) is a flavonoid compound with antitumor activity isolated from the traditional Chinese medicine Epimedium koreanum. A study in 2022 found that ICS II can inhibit the proliferation, migration, and invasion of RCC cells. This inhibition is believed to be closely related to ferroptosis. In both in vitro and in vivo experiments, ICS II demonstrated antitumor effects, specifically targeting RCC cells without affecting the viability of normal renal cells.

Mechanistically, ICS II downregulates GPX4 in a p53-independent manner, thereby triggering ferroptosis in RCC cells. Additionally, ICS II treatment leads to the upregulation of miR-324-3p, which negatively regulates GPX4 expression [41]. Experimental evidence has shown that ICS II significantly reduces the viability of RCC cells and inhibits the migration and invasion of renal cancer cells in a dose-dependent manner. ICS II treatment also increases the accumulation of Fe^2+^ and MDA, decreases GSH levels, and promotes ROS accumulation. These effects can be reversed by ferroptosis inhibitors. Furthermore, the combination of ICS II with chemotherapeutic agents, such as sunitinib and sorafenib, markedly enhances the cytotoxic effects on RCC cells, demonstrating a significant synergistic effect. This suggests a potential combined therapeutic strategy for treating RCC.

These findings suggest that ICS II may be a promising therapeutic agent for RCC. However, current research has limitations, such as validation in only two cell lines and animal models. As a promising candidate for RCC treatment, further exploration of ICS II is necessary to fully understand its potential and efficacy.

#### 4.1.2. Artesunate

Artesunate (ART) is a derivative extracted from artemisinin, a traditional Chinese medicine. It has demonstrated antitumor effects in various non-urinary system cancers [42,48,49,50]. Due to the development of drug resistance in renal cancer, traditional drugs like sunitinib have limitations. Researchers studied the effect of ART on sunitinib-resistant renal cancer cell lines Caki-1, 78-O, KTCTL-26, and A-498 [51], finding that ART inhibits the growth of three of these cell lines through mechanisms involving cell cycle arrest and regulation of cell cycle proteins. 

Notably, the effect of ART was unique in the KTCTL-26 cell line. ART induces ferroptosis in tumor cells by promoting the production of ROS and reducing glutathione (GSH) levels. Moreover, ART significantly reduced the expression of GPX4, a key component required for antioxidant defense, which is associated with ferroptosis. Additionally, p53 expression, which is closely linked to ferroptosis, was altered only in the KTCTL-26 cell line [52].

Experimental data showed that treatment with 5 μM ART in KTCTL-26 cells resulted in significant growth inhibition. Furthermore, ART at concentrations of 20 μM and 30 μM exhibited proliferation inhibition on both the parent and resistant KTCTL-26 cells. The use of the antioxidant Trolox to block free radicals significantly eliminated the proliferation-inhibitory effects of ART, further confirming that ART exerts its effects by inducing ferroptosis. These results suggest that ART has a significant inhibitory effect on the growth and survival of tumor cells by modulating ferroptosis-related molecular mechanisms, and p53 may serve as a targeted predictive marker for ART efficacy. Overall, ART may offer a promising, supplementary treatment option for patients with advanced or drug-resistant renal cancer.

#### 4.1.3. Lycorine

Lycorine, a compound isolated from plants of the Amaryllidaceae family, is commonly used in traditional Chinese medicine. It exhibits various biological activities, including antiviral, antimalarial, anti-inflammatory, and antitumor effects with relatively low side effects [53]. Despite the unclear potential targets and mechanisms of lycorine, the significant antitumor activity makes lycorine a candidate anticancer drug [54,55]. Research suggests that lycorine’s antitumor effect in renal cancer may be associated with the induction of ferroptosis [43]. Lycorine reduces the expression level of GPX4 and simultaneously increases the expression level of the key enzyme ACSL4 in lipid synthesis, promoting the increase in the lipid peroxidation marker MDA. In addition, Ferrostatin-1, as an inhibitor of ferroptosis, can reverse these phenomena caused by lycorine. Further research has discovered that lycorine exerts a significant inhibitory effect on the proliferation of RCC cells. This inhibitory action is both time- and dose-dependent, with a half-maximal inhibitory concentration (IC50) value of 5–10 μM. However, this hypothesis has limitations as it has only been validated in cell lines and lacks animal and clinical studies.

#### 4.1.4. Luteolin

Luteolin (Lut) is a natural flavonoid widely found in fruits and vegetables and has been shown to have potent anticancer activity [56]. Studies have revealed that Lut significantly inhibits the survival of renal cancer cells both in vivo and in vitro, a phenomenon accompanied by excessive intracellular ferrous ion accumulation and abnormal depletion of GSH [44]. Additionally, Lut induces mitochondrial membrane potential imbalance, classic mitochondrial ferroptosis morphological changes, ROS production, and iron-dependent lipid peroxidation in renal cancer cells. These changes induced by Lut can be partially reversed by the ferroptosis inhibitors DFO and Ferrostatin-1, indicating that Lut-treated renal cancer cells undergo ferroptosis. Mechanistically, Lut exerts its anticancer effects by excessively upregulating HO-1 expression and activating the labile iron pool (LIP), triggering ferroptosis during the treatment process. This makes Lut a promising candidate for renal cancer treatment. However, further validation across various renal cancer cell lines is needed, and whether Lut can be used as a clinical anticancer drug remains unclear, requiring extensive and large-scale clinical trials and follow-up studies.

#### 4.1.5. Salinomycin

Recent studies have shown that natural compounds and their derivatives possess antitumor effects, especially when combined with established therapies, achieving unexpected therapeutic outcomes [50,57,58]. Salinomycin (Sal) is one such natural compound, a monocarboxylic polyether ionophore antibiotic that gained attention in cancer research in 2009. A high-throughput screening of 16,000 compounds revealed its robust antitumor properties [59]. Subsequent studies confirmed Sal’s specific targeting of cancer stem cells (CSCs) and its ability to sensitize treatment-resistant cells [60]. Recent research indicates that Sal exerts anticancer effects in renal cancer by conferring sensitivity to ferroptosis. Firstly, studies found that PDIA4, the largest protein member of the PDI family, mainly functions as a molecular chaperone binding to unfolded protein substances. PDIA4 catalyzes the formation of disulfide bonds between cysteine residues, facilitating protein folding and increasing in response to intensified endoplasmic reticulum stress [61,62]. PDIA4 is upregulated in RCC, and Sal treatment downregulation of PDIA4 suppressed activating transcription factor 4 (ATF4) and its downstream protein SLC7A11, thereby inactivating GPX4 and promoting ferroptosis. Furthermore, research shows that high PDIA4 levels are associated with poorer prognosis in RCC patients, suggesting that PDIA4 could be a potential prognostic marker for RCC. Sal, meanwhile, emerges as a promising therapeutic agent for RCC.

### 4.2. Drug Resistance and Ferroptosis in Classic Therapies

#### 4.2.1. Sorafenib Resistance

Sorafenib, a multikinase inhibitor, was the first targeted drug approved by the FDA for the treatment of metastatic clear-cell renal-cell carcinoma (ccRCC) [63]. Initially, researchers believed it could act as an effective inducer of ferroptosis by inhibiting SLC7A11 activity [64]. Unfortunately, the high incidence of sorafenib resistance has emerged as a significant obstacle to its therapeutic application. Recent studies have identified the overexpression of dipeptidyl peptidase 9 (DPP9) as a contributor to sorafenib resistance [32]. DPP9 is a member of the dipeptidyl peptidase family that can cleave dipeptides at the N-terminal proline site. Its expression is elevated in various cancers and is associated with poor clinical prognosis and chemotherapy resistance [65].

By analyzing publicly available TCGA datasets, researchers found that DPP9 mRNA expression is significantly upregulated in ccRCC tissues compared to normal renal tissues. Affinity purification and mass spectrometry confirmed the presence of Kelch-like ECH-associated protein 1 (KEAP1) in the DPP9 protein complex, indicating a protein-level interaction between DPP9 and KEAP1. KEAP1 interacts with the conserved KEAP1 binding motif ESGE in the DPP9 protein sequence, and DPP9 competitively binds to KEAP1, blocking KEAP1-mediated ubiquitination and degradation of nuclear factor erythroid 2–related factor 2 (NRF2) [66]. This interaction promotes NRF2-mediated oxidative stress pathways, reducing intracellular ROS levels.

Further experiments revealed that the overexpression of DPP9 blocks NRF2 ubiquitination and degradation, leading to upregulated SLC7A11 protein levels, which in turn causes resistance to ferroptosis in tumor cells. This mechanism is in direct contrast to the previously mentioned mechanism where sorafenib induces ferroptosis by inhibiting SLC7A11 activity, thus providing new theoretical insights into sorafenib resistance. Subsequent in vivo animal experiments and organoid models validated that knocking out DPP9 can partially reverse sorafenib resistance.

Therefore, for patients with DPP9-overexpressing resistant ccRCC, it might be possible to attempt reversing targeted therapy resistance by using small-molecule compounds to block the interaction between DPP9 and KEAP1, thereby inhibiting the excessive activation of NRF2 signaling.

#### 4.2.2. Sunitinib Resistance

In recent years, tyrosine kinase inhibitors (TKIs) and immune checkpoint inhibitors have become the preferred treatments for inoperable renal cancer. However, these treatments still face significant limitations. While molecular targeted therapies have improved outcomes for some patients, most eventually develop resistance, and sunitinib is no exception. For patients resistant to sunitinib, clinical treatment options are quite limited, presenting challenges in the clinical management of renal cancer [67].

Recent studies have identified the Absent in Melanoma 2 (AIM2) inflammasome as a novel biomarker for renal cancer, promoting cancer progression and contributing to sunitinib resistance [68]. Further research has shown that AIM2 is involved in iron ion homeostasis and response, a critical component of ferroptosis. AIM2 is closely related to the ferroptosis-associated gene ACSL4, which promotes the formation of phospholipids containing polyunsaturated fatty acids (PUFAs), thereby inducing ferroptosis [69].

Mechanistic exploration has revealed that the AIM2 inflammasome promotes the phosphorylation and proteasomal degradation of Forkhead Box O3a (FOXO3a), thereby reducing its transcriptional effect on ACSL4 and inhibiting ferroptosis. This mechanism has been observed in both resistant and normal renal cancer cell lines, indicating that AIM2 contributes to sunitinib resistance by this pathway. These findings suggest that targeting ferroptosis and the FOXO3a-ACSL4 axis might offer a new approach to overcoming sunitinib resistance in renal cancer treatment.

These findings reveal that the inhibition of ferroptosis in renal cancer cells is involved in the mechanism of resistance to conventional chemotherapeutic drugs, providing inspiration for treating patients with chemotherapy resistance: overcoming resistance and enhancing the efficacy of chemotherapy by targeting and inducing ferroptosis.

### 4.3. Combination Therapy with Ferroptosis Inducers

#### 4.3.1. Everolimus Combined with Ferroptosis Inducers Can Overcome Sorafenib and Sunitinib Resistance

Everolimus, approved by the FDA as a mammalian target of rapamycin (mTOR) inhibitor, is used as a second-line treatment for metastatic renal-cell carcinoma (RCC) resistant to sorafenib or sunitinib. However, the efficacy of Everolimus has also been hampered by resistance. Erastin and RSL3 are classic ferroptosis inducers that induce ferroptosis by inhibiting the cystine/glutamate antiporter system Xc− and GPX4, respectively [21].

Recently, studies have explored the impact of the combination of Everolimus with RSL3 or Erastin on RCC cells [45]. Experimental data indicate that this drug combination can significantly increase the mortality rate of RCC cells, demonstrating a pronounced synergistic lethal effect compared to the use of individual drugs. By employing a variety of cell death inhibitors, the study attributes the observed cell death primarily to ferroptosis. Furthermore, the cell death induced by this combined treatment in RCC is characterized by a decrease in glutathione (GSH) levels, an increase in reactive oxygen species (ROS) production, and elevated levels of lipid peroxides. Additional mechanistic research reveals that Everolimus reduces the expression level of glutathione peroxidase 4 (GPX4) by inhibiting the mTOR signaling pathway, which corresponds with the dependency of GPX4 synthesis on the activation of mTOR [70].

This combination presents a promising therapeutic option for patients with resistant renal cancer and helps overcome the clinical limitations of Everolimus.

#### 4.3.2. URB597

Studies have demonstrated the potentiated antitumor efficacy of combining RSL3, a ferroptosis inducer, with URB597, a selective FAAH inhibitor, in renal cancer cells [46]. FAAH, a member of the serine hydrolase family, was initially identified as a major catabolic enzyme that regulates various metabolic pathways and pathophysiological processes, including cancer cell proliferation. FAAH expression is upregulated in multiple tumor tissues, and its inhibitors have demonstrated anti-invasive and anti-metastatic effects on various cancer cells. Some FAAH inhibitors have shown synergistic effects with chemotherapeutic drugs [71,72], which has become a research hotspot in cancer treatment, especially since clinical trials have shown that FAAH inhibitors have good tolerability and minimal toxicity [73].

The combination therapy reduces renal-cell carcinoma (RCC) cell viability. This effect, more pronounced than with single-agent treatments, is characterized by the induction of ferroptosis and G1 cell cycle arrest. The synergistic mechanism likely involves the modulation of lipid metabolism and an increase in cellular ROS, hallmarks of ferroptosis. RNA sequencing analysis indicates that the combined treatment more significantly regulates genes associated with cell proliferation, cell cycle, migration, and invasion, as well as those involved in ferroptosis, compared to individual treatments. This regulation suggests a coordinated impact on biological processes that are crucial for RCC progression. 

Furthermore, the combination treatment affects the PI3K-AKT signaling pathway, known for its role in cell survival and growth, and may enhance the sensitivity of RCC to ferroptosis induction. Targeting this pathway could be a strategic approach to augment the therapeutic efficacy of ferroptosis inducers.

In conclusion, the synergistic approach of combining URB597 and RSL3 presents a therapeutic strategy for RCC, harnessing the endocannabinoid system and ferroptosis pathways with the potential to target the PI3K-AKT pathway. This strategy may pave the way for developing more effective treatments for renal cancer, particularly for patients with advanced stages where existing therapies have shown limited efficacy.

### 4.4. Novel Synthetic Materials

#### MIL-101(Fe)@RSL3

MIL-101(Fe) nanoparticles loaded with RSL3 form a new ferroptosis activator, MIL-101(Fe)@RSL3. By utilizing iron-rich MIL-101(Fe) nanoparticles for targeted delivery and responsive release of the ferroptosis inducer RSL3, MIL-101(Fe)@RSL3 exhibits high encapsulation efficiency and tumor-targeted delivery of RSL3. This combination triggers a cascade of ferroptosis treatment in clear-cell renal-cell carcinoma (ccRCC) [47]. In the acidic tumor microenvironment, the gradually degrading MIL-101(Fe)@RSL3 releases iron ions and RSL3, which promotes hydroxyl radical (·OH) production through the Fenton reaction. These radicals attack polyunsaturated fatty acids (PUFAs), leading to the abnormal accumulation of lipid peroxides (L-OOH). Ferrous ions further catalyze the irreversible transformation of L-OOH into highly active lipid alkoxy radicals (L-O·), initiating a cascade of ferroptosis. Additionally, RSL3 directly inhibits GPX4′s detoxification of L-OOH. Unlike the limited antitumor effects of free RSL3, MIL-101(Fe)@RSL3 has a high encapsulation rate (88.7%), resulting in significantly enhanced antitumor effects and abnormal PUFA metabolism in renal cancer. Notably, experiments have confirmed that this novel synthetic nanomaterial does not exhibit noticeable cytotoxicity to normal cells even at high concentrations, making it a promising drug for renal cancer treatment.

### 4.5. The Combination of Ferroptosis Therapy and Immunotherapy

#### iRGDbcc-USINP

Iron nanoparticles, such as MIL-101(Fe), as well as widely used iron oxide nanoparticles (IONPs) and iron–organic frameworks for cancer diagnosis and therapy [74,75], induce ferroptosis through catalyzing the Fenton reaction and accelerating reactive oxygen species (ROS) production for cancer-specific treatment [76], They can also activate immune responses by releasing damage-associated molecular patterns (DAMPs) during ferroptosis, contributing to cancer immunotherapy. Most iron nanomaterials rely on the release of ferrous ions to trigger the Fenton reaction [77]. However, due to the low ROS conversion efficiency of ferrous ions in the tumor microenvironment (pH 5.5–6.5), these nanoparticles often require synergistic action with other components or combined therapies. For example, FDA-approved Ferumoxytol shows certain antitumor immunotherapy effects via the Fenton reaction but requires a high dosage [78].

Recent studies have synthesized ultra-small single-crystal Fe nanoparticles (bcc-USINPs) composed of a 2.3 ± 0.2 nm bcc zero-valent iron core and a 0.7 ± 0.1 nm crystalline Fe_3_O_4_ shell [79]. Under normal physiological conditions, the ultra-thin iron oxide shell protects Fe(0) from oxidation, masking its activity. In the acidic tumor environment, the iron oxide shell corrodes and breaks, exposing Fe(0), which effectively induces ferroptosis in tumor cells through the Fenton reaction. After modification with iRGD peptides, these nanoparticles exhibit high tumor accumulation and retention. Moreover, iRGD-bcc-USINP treatment effectively induces immunogenic cell death (ICD), promotes dendritic cell (DC) maturation, and triggers adaptive T-cell responses, combining with programmed death-ligand 1 (PD-L1) immune checkpoint blockade for immunotherapy. iRGD-bcc-USINP-mediated ferroptosis therapy significantly enhances immune response rates and establishes strong immune memory. Compared to other iron-based therapeutic systems that typically require high-dose injections or combination with other ferroptosis inducers to effectively treat cancer, bcc-USINPs can induce tumor cell ferroptosis and ICD at extremely low concentrations. Their ultra-small size enables rapid renal clearance, providing a simple, safe, effective, and tumor-responsive Fe(0) delivery system for ferroptosis-based immunotherapy.

Although this study has not been validated in a renal cancer model, it has shown significant therapeutic effects in liver cancer, colon cancer, and breast cancer models. This method, which combines the induction of ferroptosis with immunotherapy, provides a new perspective on cancer treatment, and the high renal clearance efficiency of bcc-USINPs provides safety assurance for its clinical application. However, it should be pointed out that before these nano-drugs can be transformed into actual therapeutic methods, they need to go through more preclinical and clinical studies to verify their universality, clinical efficacy, and safety. Although the current research is promising, it still needs to explore key issues further, such as long-term application toxicity, stability, and cost-effectiveness.

## 5. Ferroptosis-Related Drugs and Clinical Trials

Beyond inspiring drugs and strategies aiming at renal cancer, we expanded our scope to more ferroptosis drugs applied clinically. These drugs, through diverse molecular mechanisms, have shown the ability to either promote or inhibit ferroptosis in various disease models. They have also been applied in clinical trials for the treatment of renal cancer (Table 2), providing anchor points for further studies and confirming the critical role and therapeutic potential of ferroptosis in the progression of renal cancer.

Sorafenib, widely recognized as a ferroptosis inducer, is particularly important in renal cancer treatment due to its diverse mechanisms of action (Table 2). Numerous clinical trials have confirmed sorafenib’s central role in treating renal cancer (Table 2). However, the issue of chemotherapy resistance persists throughout treatment, and inducing ferroptosis offers new hope for mitigating this challenge. By targeting ferroptosis in combination with other drugs, a new strategy to overcome resistance during sorafenib treatment has emerged [80,81].

Apart from sorafenib, other drugs also demonstrate notable mechanisms in ferroptosis. The role of HIF-1α/SLC7A11 in liver fibrosis and the potential impact of METTL14/m6A/FTH1 in cervical cancer provide new insights for developing innovative cancer treatment strategies. Ongoing clinical trials, such as the application of Deferoxamine in locally advanced or metastatic renal cancer, have demonstrated its safety and efficacy in clinical practice [82]. This reinforces our confidence in the clinical application of ferroptosis inducers.

While the study of ferroptosis inducers shows advantages and innovation, we also recognize the existing limitations. The specificity and selectivity of ferroptosis inducers require further investigation to ensure they effectively target tumor cells while minimizing effects on normal cells. Additionally, the understanding of the mechanisms of many compounds remains incomplete, limiting our ability to predict their behavior in complex biological systems. Therefore, despite the enormous potential observed in the field of ferroptosis inducers, more in-depth quantitative and systematic research regarding drug mechanisms, safety, and efficacy in treating renal cancer remains necessary to realize a more responsible and extensive application of these drugs.
cancers-16-03131-t002_Table 2Table 2Status of clinical studies of ferroptosis drugs up until the completion of the review manuscript.AgentEffect on FerroptosisProposed MechanismDisease ModelRefs.IndicationNCTStatus

HIF-1α/SLC7A11Liver fibrosis[83]Metastatic Renal-Cell CarcinomaNCT01982097CompletedNCT00073307Completed

ERK-TRIM54/FSP1; SLC7A11; HBXIP/SCD axis; GABARAPL1; QSOX1/NRF2HCC[84,85,86,87,88]Renal CancerNCT01557127CompletedSorafenibInducerSLC7A11Nasopharyngeal carcinoma[89]Renal-Cell CarcinomaNCT00661375CompletedNCT00678288TerminatedNCT00618982CompletedNCT00586105Completed

METTL14/m6A/FTH1Cervical cancer[90]Advanced Renal-Cell CarcinomaNCT01508364CompletedNCT00895674CompletedNCT00771147Completed




Elderly mRCCNCT01728948CompletedTemozolomideInduceriron metabolism; DMT1Glioblastoma[91]Hereditary Leiomyomatosis and Renal-Cell CancerNCT04603365WithdrawnStatinsInducerTarget CoQ/FSP1Triple-negative breast cancer[92]Renal-Cell CarcinomaNCT00490698CompletedDexamethasoneInducerDPEP1;GSHHT1080[93]Metastatic Renal-Cell CarcinomaNCT00176280WithdrawnDeferoxamineInhibitorIron ionstraumatic spinal cord injury[82]Locally Advanced or Metastatic Renal-Cell CarcinomaNCT04006522Recruiting




Clear-Cell RCCNCT06090331AvailableSeleniumInhibitorTFAP2c, Sp1,GPX4Stroke[94]Clear-Cell RCCNCT02535533Active, not recruiting





NCT05363631Recruiting


## 6. Conclusions

Ferroptosis is closely associated with disruptions in iron metabolism, lipid metabolism, and the antioxidant system. Renal cancer cells are prone to iron accumulation, as well as abnormalities in amino acid and lipid metabolism. Herbal medicines and natural compounds, as activators of ferroptosis, provide new potential drugs for developing further therapeutic strategies for renal cancer. Ferroptosis inducers have also been found to counteract the development of resistance when used in combination with classical drugs for renal cancer treatment. Additionally, we emphasize the potential of combining ferroptosis therapy with immunotherapy, as well as the application of new nanomaterials in targeting iron overload in tumor cells. Existing clinical trial results highlight both the feasibility and challenges of translating ferroptosis inducers into clinical applications, underscoring the need for further research on these compounds.

While progressing research on ferroptosis inducers is promising, we also recognize the existing limitations and challenges. Future studies need to explore in greater depth the specificity, selectivity, safety, and stability of ferroptosis inducers for long-term application.

## Figures and Tables

**Figure 1 cancers-16-03131-f001:**
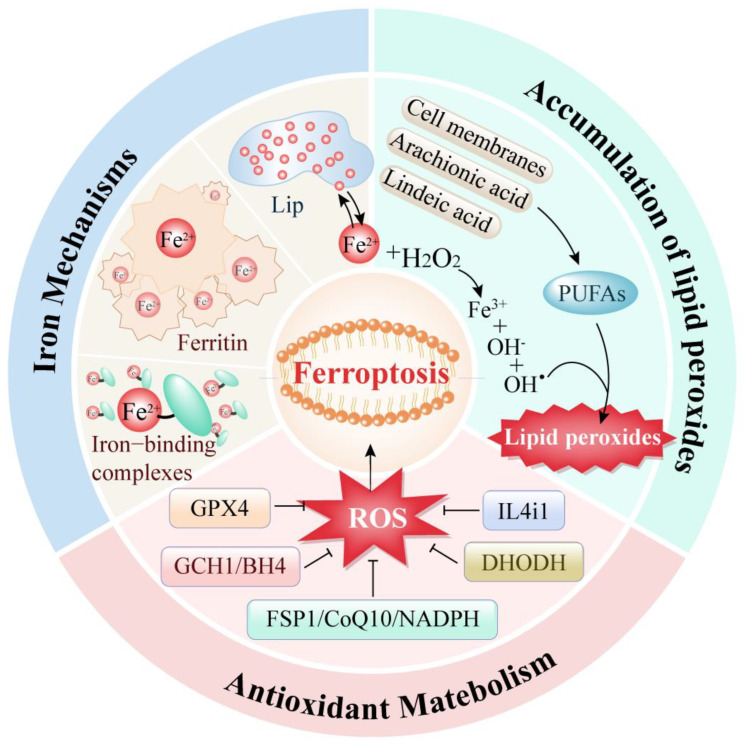
Three regulatory pathways of ferroptosis.

**Figure 2 cancers-16-03131-f002:**
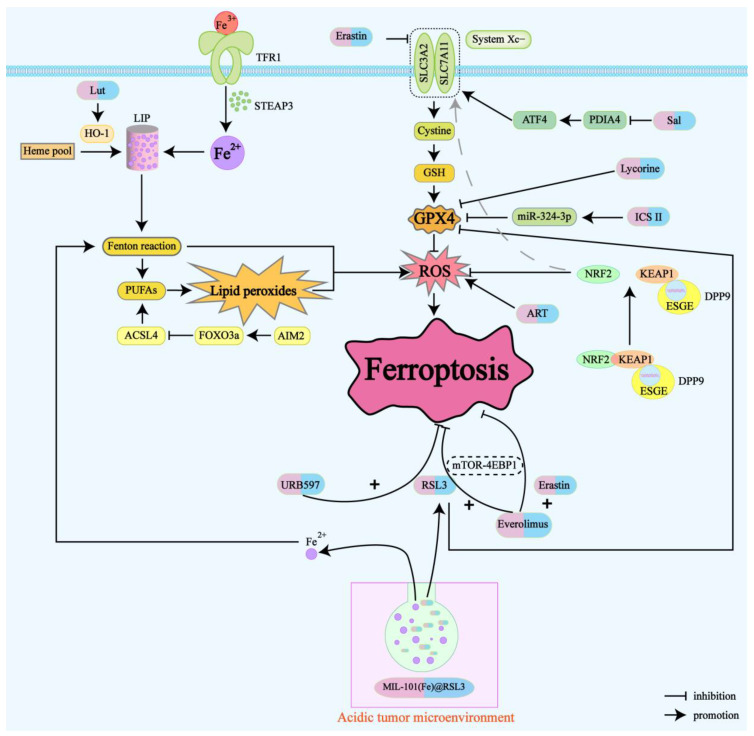
Drug mechanism map of renal cancer treatment based on ferroptosis. Drug-induced ferroptosis mechanism: ICS II, lycorine, and Sal target GPX4 followed by ART-induced ROS generation; Lut upregulates the expression of HO-1 and activating LIP; and Everolimus combined with RSL3/Erastin inhibits the mTOR-4EBP1 axis; Chemotherapy resistance: decreased NRF2 ubiquitination and degradation and suppressed ferroptosis leads to sorafenib resistance; AIM 2 inflammasome reduces transcription of ACSL 4 and suppresses ferroptosis, leading to sunitinib resistance.

**Table 1 cancers-16-03131-t001:** Overview of researched drugs targeting ferroptosis in treating renal cancer.

Type of Drugs	Drugs	Function	Experimental Object	Targeting	Refs.
Extract of Traditional Chinese Medicine	Icariin II	Promote ferroptosis	Human RCC (ACHN, A498, 786-O, Caki-1) male BALB/c nude mice	miR-324-3p, GPX4	[41]
	Artesunate	Promote ferroptosis	Parental and sunitinib-resistant KTCTL-26	ROS, GPX4, p53	[42]
	Lycorine	Promote ferroptosis	Human RCC (786-O, A498 and Caki-1); HK-2	GPX4, ACSL4	[43]
	Luteolin	Promote ferroptosis	Human ccRCC (786-O and OS-RC-2); HK-2; OS-RC-2 cells injected BALB/c nude mice	HO-1, LIP	[44]
Natural Compound	Salinomycin	Promote ferroptosis	HK-2; 786-O, 769-P, ACHN, CAKi-1; 786-O cells injected BALB/c nude mice; Human ccRCC specimens	PDIA4-ATF4-SLC7A11-GPX4 axis	[31]
Drugs Combined with Ferroptosis Inducers	Everolimus	Promote ferroptosis	RCC cell line (ACHN, Caki-1), HEK-293	mTOR-4EBP1 axis	[45]
	URB597	Promote ferroptosis	Kidney 293T cells RCC cell line (Caki-1, 786-O, OS-RC-2, SW839, GRC-1 G-401 c A498) Patient tumor tissue samples	PI3K-AKT signaling pathway, ROS, GPX4	[46]
Novel Synthetic Materials	MIL-101(Fe)@RSL3	Promote ferroptosis	786-O, HK-2, LO2, injected female BALB/C nude mice with 4 × 10^6^ 786-O cells	GPX4, Fenton reaction	[47]

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
