# Peer review of "Ferroptosis in Renal Cancer Therapy: A Narrative Review of Drug Candidates"

_cancers, 2024, doi:10.3390/cancers16183131_

Round 1

Reviewer 1 Report

Comments and Suggestions for Authors

Ferroptosis is a type of programmed cell death dependent on iron and characterized by the accumulation of lipid peroxides, and is genetically and biochemically distinct from other forms of regulated cell death such as apoptosis. Here, the authors prepared the narrative review on the ferroptosis mechanisms to be considered for clinical implementation in renal cancer (RCC) disease. Firstly, a large chapter was included on traditional Chinese medicine and natural compounds in RCC. Then, a classic approach with combination therapy with ferroptosis inducers was added. A final minor comment is that adding a table summarizing available/ongoing clinical trials on the above mentioned therapies would be advisable. 

Author Response

Comments 1: A final minor comment is that adding a table summarizing available/ongoing clinical trials on the above mentioned therapies would be advisable.

Response 1: 

    Thank you for your valuable and thoughtful suggestion. We have revisited the entire text and found that the research on the drugs initially mentioned has not yet progressed to the clinical trial stage. To better reflect the feasibility of the therapy, we have created Table 1, which reflects the feasibility of the treatment from a mechanistic perspective. Additionally, we have actively sought other ferroptosis-related drugs and found that some of these drugs have been used in clinical trials for the treatment of renal cancer. We have summarized these drugs and clinical trials in Table 2.

    Please see lines 186-187, 480-481 for further details.

Reviewer 2 Report

Comments and Suggestions for Authors

Title = should mention that this is a narrative review - Minor

Abstract - nicely depicting the essence of  the article - No Remarks

Introduction - a comprehensive review on the current status of the literature, strong foundation for next paragraphs

Mechanisms of Ferroptosis Regulation - in-depth paragraph on the basis of ferroptosis in RCC - No remarks   Ferroptosis and renal cancer - rows 138 - 157 - lack of enough references - Minor    

Author Response

Comments 1: Title = should mention that this is a narrative review - Minor

Response 1: 

     Thank you for your insightful comments. We appreciate the suggestion to clarify that the manuscript is a narrative review in the title. We have acknowledged that the original title, "Research Progress on Ferroptosis-Based Drugs for Renal Cancer Treatment," may not sufficiently convey the nature of the review and the breadth of the article's coverage. To better reflect the content of the paper and highlight its narrative review aspect, we have revised the title accordingly. The title has now been updated to: "Ferroptosis in Renal Cancer Therapy: A Narrative Review of Drug Candidates".

    Please see lines 2-3 for further details.

Comments 2: Abstract - nicely depicting the essence of  the article - No Remarks

Response 2: Thanks for your comments.

Comments 3:  Introduction - a comprehensive review on the current status of the literature, strong foundation for next paragraphs

Response 3: Many thanks for your comments on Introduction section.

Comments 4: Mechanisms of Ferroptosis Regulation - in-depth paragraph on the basis of ferroptosis in RCC - No remarks

Response 4:  Thanks for your kind comments.

Comments 5: Ferroptosis and renal cancer - rows 138 - 157 - lack of enough references - Minor

Response 5:  

    We would like to express my sincere gratitude for your valuable suggestions and meticulous review. Following your feedback, we have incorporated several latest references([28]-[37], [39], [40]) into the manuscript to enhance the academic foundation and depth of our argument.

    Please see lines 139-160 for further details.

Reviewer 3 Report

Comments and Suggestions for Authors

This review article, “Research Progress on Ferroptosis-Based Drugs for Renal Cancer Treatment” by Lingyan Yu et al., represents the mechanism of different ferroptosis-related drugs to treat renal cancer. The information in this manuscript does not focus on preclinical and clinical status. The explanation of the functional implementation of individual therapeutic payloads is missing; there is a lack of updated statistical information in the introduction. Image diagrams lack mechanistic explanation. Individual drug molecule inhibition potency is not presented quantitatively. The review article does not cover a major portion of the research community. The present standard of the article does not meet the quality of the journal cancers. It is not recommended for publication in cancers.   

Author Response

Comments 1: The information in this manuscript does not focus on preclinical and clinical status.

Response 1:

Thank you for your insightful comments. We have added one new section on clinical trials (Section 5, page 13), providing a brief summary of clinical trials for the treatment of renal cancer with ferroptosis-related drugs, and organizing the drugs and clinical trials in Table 2.

Please see lines 450-481 for further details.

Comments 2: The explanation of the functional implementation of individual therapeutic payloads is missing

Response 2:

Thanks to the review for raising this key issue, we added Table 1 to the manuscript to summarize the targets and functions of the drug action, and also added interpretation of individual therapeutic functions to the content of the manuscript.

Please see lines 186-187, 198-205, 218-223, 242-244,381-390 for further details.

Comments 3: There is a lack of updated statistical information in the introduction.

Response 3:

We appreciate you for bringing attention to this significant matter. We have checked the literature carefully and updated the latest statistics in the revised manuscript.

Please see lines 40-41 for further details.

Comments 4: Image diagrams lack mechanistic explanation.

Response 4:

We express our gratitude for your important advice. In accordance with the reviewer's suggestion, we have added an explanation of the mechanism below the Image diagrams.

Please see lines 66-68, and 178-183 for further details.

Comments 5: Individual drug molecule inhibition potency is not presented quantitatively.

Response 5:

We would like to express my sincere gratitude for your valuable suggestions and meticulous review. Due to the differences in each drug research method and experimental subjects, there are problems in quantitative presentation because there is no universal analytical index. But we are also adding some experimental content behind some drugs, hoping to more intuitively reflect the effect of drugs to inhibit renal cancer through ferroptosis. In addition, most drugs have been shown in other studies to interfere with the progression of tumors or diseases by affecting ferroptosis.

Please see lines 224-230, and 244-248 for further details.

Round 2

Reviewer 3 Report

Comments and Suggestions for Authors

The manuscript has been revised and upgraded to assist readers and is now suitable for publication.